# Proportional Fairness in Clustering:
# A Social Choice Perspective

**Leon Kellerhals**
Technische Universität Clausthal
`leon.kellerhals@tu-clausthal.de`

**Jannik Peters**
National University of Singapore
`peters@nus.edu.sg`

## Abstract

We study the proportional clustering problem of Chen et al. (ICML'19) and relate it to the area of multiwinner voting in computational social choice. We show that any clustering satisfying a weak proportionality notion of Brill and Peters (EC'23) simultaneously obtains the best known approximations to the proportional fairness notion of Chen et al., but also to individual fairness (Jung et al., FORC'20) and the "core" (Li et al., ICML'21). In fact, we show that any approximation to proportional fairness is also an approximation to individual fairness and vice versa. Finally, we also study stronger notions of proportional representation, in which deviations do not only happen to single, but multiple candidate centers, and show that stronger proportionality notions of Brill and Peters imply approximations to these stronger guarantees.

## 1 Fair clustering

Fair decision-making is a crucial research area in artificial intelligence and machine learning. To ensure fairness, a plethora of different fairness notions, algorithms and settings have been introduced, studied, and implemented. One area in which fairness has been applied extensively is *(centroid) clustering*: We are given a set of $n$ data points which we want to partition into $k$ clusters by choosing $k$ "centers" and assigning each point to a center by which it is *represented well*. Fairness now comes into play when, e.g., the data points correspond to human individuals.

Fairness notions in clustering usually depend on one decision: whether one takes demographic information (such as gender, income, etc.) into account or whether one is agnostic to it. A large part of work on fair clustering has focused on incorporating such demographic information, starting with the seminal work of Chierichetti et al. [2017] who aimed to proportionally balance the number of people of a certain type in each cluster center. However, not all work on fair clustering relies on demographic information. Independently, and in different contexts, Jung, Kannan, and Lutz [2020] and Chen, Fain, Lyu, and Munagala [2019] instead tried to derive fairness notions from the instance itself. For Jung et al. this lead to their notion of *individual fairness*: Given a population of size $n$, with $k$ cluster centers to be opened, every agent should be entitled to a cluster center not further away than their $\frac{n}{k}$-th neighbor. While this is not always achievable, Jung et al. gave a simple algorithm achieving a 2-approximation to this notion. Chen et al. were motivated not by being fair towards individual members of the population (or agents), but towards groups of agents, defining their notion of *proportional fairness*: no group of size at least $\frac{n}{k}$ should be able to suggest a cluster center they all would be better off with. This notion is also not always achievable, and Chen et al. gave a simple $(1 + \sqrt{2})$-approximation for it.

38th Conference on Neural Information Processing Systems (NeurIPS 2024).

So far, the individual and proportional fairness notions (and some other related fairness notions) have existed in parallel, with similarities between the two being acknowledged but not formalized.[1] In their survey, Dickerson et al. [2023b] highlight this as a general issue in fair clustering: "each notion that was introduced [...] does not refer to or consider the interaction with the previously introduced fairness notions in clustering". Moreover, they call for "other fairness notions in clustering that are also compatible with one another" and "general notions which possibly encompass existing ones".

We follow this call and prove proportional and individual fairness, as well as a fairness notion by Li et al. [2021] which we will call the *transferable core*, to be tightly related to another. In an effort to encompass these three notions, we make use of proportionality axioms from *multiwinner voting*, an area in computational social choice [Lackner and Skowron, 2022]. Here, given the votes of $n$ agents, the goal is to elect a size-$k$ committee which fulfills some proportionality guarantee. We lift one of the simplest proportionality guarantees (JR) to work with metric distances and prove that any clustering fulfilling our guarantee also fulfills the best approximations for the three notions, all *simultaneously*. Moreover, such a clustering can be computed in polynomial time. Taking the multiwinner voting approach further, we also look at the lifted version of a stronger proportionality guarantee (PJR). This changes how points (agents) interact with cluster centers as they become represented not by one, but possibly multiple centers. While this is not standard for "vanilla" clustering, it is very fitting for more democratic settings, where the chosen "centers" end up possessing voting power to represent the agents. The resulting proportionality guarantee indeed highly relates to work by Ebadian and Micha [2024] who, motivated by *sortition* (the randomized selection of citizens' panels [Flanigan et al., 2021]), introduced a generalization of the proportional fairness notion. Indeed, the multiwinner voting perspective allows us to prove better approximation guarantees for their fairness notion.

**Our contributions.** As our first main result, we provide a simple bridge between proportional fairness and individual fairness (see Section 2). Any approximation of the former is also an approximation of the latter. In particular, for any $\alpha, \beta \geq 1$ we show that (i) any $\alpha$-approximation to proportional fairness is also an $(1 + \alpha)$-approximation to individual fairness and (ii) any $\beta$-approximation to individual fairness is also a $2\beta$-approximation to proportional fairness. These approximations are tight. We also prove a similar connection between proportional fairness and the transferable core. Our connections imply for instance that bi-criteria approximations that optimize $k$-means and, say, individual fairness [Vakilian and Yalçıner, 2022, Bateni et al., 2024] also maintain approximations guarantees to the other fairness notions. Further, if one wants to show incompatibility of a different clustering notion with approximate proportional or individual fairness, it is sufficient to show this for one of the two notions, instead of creating instances for both (as done by Dickerson et al. [2023a]).

Secondly, in Section 3, we draw a connection to the area of multiwinner voting and reinterpret proportionality notions introduced by Brill and Peters [2023] to work with distance metrics; we call the resulting guarantees mJR and mPJR. Both of these are efficiently computable when the space of possible centers is finite. Remarkably, with simple proofs, we are able to show that any clustering satisfying mJR achieves the *best known* approximations to individual and proportional fairness notions and the transferable core. For the transferable core, we even improve upon the bound derived by Li et al. [2021]. Finally, motivated by settings such as sortition and multiwinner voting in which agents do not only care about their closest cluster center but are represented by multiple centers, we show that a strong core stability guarantee (introduced by Ebadian and Micha [2024]) can be achieved by any clustering satisfying mPJR. We also deal with the case in which the center candidate space is unbounded (e.g., in Euclidean clustering settings), in which the above-mentioned algorithms can become intractable. Here, we show that satisfying the proportionality guarantees only for the set of agents is sufficient to obtain constant-factor approximations to proportional fairness and the core stability guarantee by Ebadian and Micha [2024].

Lastly, in Section 4, we focus on sortition: Here, the set of agents and cluster candidates is equal and each agent must be chosen with equal probability. Employing techniques from the above results, we are able to give a simpler proof achieving a better approximation guarantee for the core notion by Ebadian and Micha [2024].

Figure 1 (left) gives an overview over our results and our achieved approximation guarantees. Proofs of some results are deferred to a full version of this manuscript [Kellerhals and Peters, 2023].

---

[1]For instance, in a recent tutorial on fair clustering [Awasthi et al., 2022], the two notions were treated as separate unconnected paradigms.

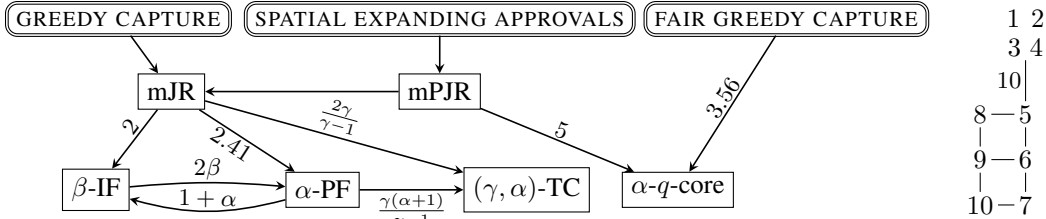

Figure 1: *Left:* An overview over connections between and bounds on fairness notions, i.e., $\alpha$-proportional fairness ($\alpha$-PF), $\beta$-individual fairness ($\beta$-IF), the $(\gamma, \alpha)$-transferable core ($(\gamma, \alpha)$-TC), and the $\alpha$-$q$-core. See Sections 2 and 3 for the corresponding definitions and results. If A $\to \Pi$, then algorithm A produces outcomes satisfying $\Pi$. If $\Pi \to \Gamma$, then any outcome satisfying $\Pi$ also satisfies $\Gamma$. If $\Gamma$ takes a parameter $\alpha$, then the label specifies the parameter that can be satisfied (for the transferable core, the result holds for all $\gamma > 1$). *Right:* The metric space for the examples used throughout the paper. Edges without labels have length 1, the distance between any two points is given by the length of the shortest path between them.

**Related work.** *Individual fairness* was introduced by Jung et al. [2020]. Since then, follow-up work mainly focused on bi-criteria approximation guarantees [Mahabadi and Vakilian, 2020, Negahbani and Chakrabarty, 2021, Vakilian and Yalçıner, 2022, Chhaya et al., 2022, Bateni et al., 2024]. Additionally, Han et al. [2023] studied individual fairness for clustering with outliers and Sternbach and Cohen [2023] incorporated demographic information into individual fairness. The individual fairness notion was also carried over to the setting of approval-based multiwinner voting [Brill et al., 2024]. We mention that the name "individual fairness" is also used for other (unrelated) fairness notions [e.g. Kar et al., 2023, Chakrabarti et al., 2022].

*Proportional fairness* was first studied by Chen et al. [2019]. Micha and Shah [2020] showed that the GREEDY CAPTURE algorithm by Chen et al. achieves better approximation guarantees in certain metric spaces (including the Euclidean space with the 2-norm) and studied its complexity. Li et al. [2021] introduced notions inspired by Chen et al., which are related to the transferable core concept from algorithmic game theory. Aziz et al. [2024] introduced proportionality axioms and rules directly inspired from social choice theory to proportional clustering. Among other things, they showed that every outcome satisfying DPRF (see Section 3.1) achieves an $\left(1 + \sqrt{2}\right)$-approximation to proportional fairness. Further connections to social choice or relations between the above fairness notions of Jung et al. [2020] or Li et al. [2021] remain unexplored, though.

Ebadian and Micha [2024] study proportionality in the setting of *sortition* (see e.g., Flanigan et al. [2021]), proposing a generalization of proportional fairness and a refined variant of GREEDY CAPTURE. This variant and its proportionality were used by Caragiannis et al. [2024a] to construct panels whose decisions align with that of the underlying population. The most recent work directly related to ours was created independently and in parallel to ours by Kalaycı et al. [2024]. They study proportional fairness and the transferable core in an incomplete information setting and show that just knowing the order of the distances to between agents and center candidates suffices to achieve a 5.71-approximation to proportional fairness.

Caragiannis et al. [2024b] study proportional fairness in a *non-centroid* based fair clustering setting, in which points are not assigned to cluster centers. Instead they are grouped into clusters and derive utility based on the other agents in their cluster. For this setting, Caragiannis et al. [2024b] also build on the proportional fairness framework studied in this work and also take inspiration from concepts from multiwinner voting: in their case, they adopt the *FJR* axiom of Peters et al. [2021]. The setting of non-centroid clustering is closely related to the study of *hedonic games*, with the difference being that in hedonic games, the number of clusters is not pre-determined. See for instance, Fanelli et al. [2021], Demeulemeester and Peters [2023], and Fioravanti et al. [2023] for recent works on approximate core stability in hedonic games. Ahmadi et al. [2022], Aamand et al. [2023], and Mosenzon and Vakilian [2024] further studied a notion of individual stability for clustering, in which individual agents should not be able to deviate from their clusters. This, however, is unrelated to group stability as studied in this work.

*Multiwinner voting* is the branch of computational social choice theory dealing with selecting multiple instead of just one candidate as a winner. A main branch herein focuses on *proportionality*. While much of the literature on proportionality, starting with Aziz et al. [2017], focuses on approval preferences (see Lackner and Skowron [2022] for a recent book on this topic), proportionality notions also exist for ordinal preferences [Dummett, 1984]. These notions were recently strengthened by Aziz and Lee [2020, 2021] and Brill and Peters [2023], with the latter forming the basis for the proportionality axioms we discuss in this paper. We are further closely related to the works of Caragiannis et al. [2022] and Ebadian et al. [2022] who studied the representation of a given committee by investigating the distances of agents to their $q$-closest committee member.

**Model and notation.** Let $(\mathcal{X}, d)$ be a (pseudo)-metric space with a distance function $d \colon \mathcal{X} \times \mathcal{X} \to \mathbb{R}$ satisfying $d(i, i) = 0$, $d(i, j) = d(j, i)$ and $d(i, j) + d(j, k) \geq d(i, k)$. Let $i \in \mathcal{X}$ be a point. For $r \in \mathbb{R}$, define $B(i, r) = \{j \in \mathcal{X} \colon d(i, j) \leq r\}$ to be the ball of radius $r$ around $i$. For $W \subseteq \mathcal{X}$, let $d(i, W) = \min_{c \in W} d(i, c)$. For $q \leq |W|$, $d^q(i, W)$ is distance to the $q$-th closest point in $W$ to $i$. Note that $d^1(i, W) = d(i, W)$ and that $d^q(i, W) \leq d(i, j) + d^q(j, W)$ for $i, j \in N$.

Throughout the paper, we are given a set of *agents* $N = [n]$ and a (possibly infinite) set of *candidates (facilities)* $C$, both of which lie in a metric space $(\mathcal{X}, d)$, and a number $k \in \mathbb{N}^+$. A *clustering* or *outcome* is a subset $W \subseteq C$ of at most $k$ candidates. The elements $c \in W$ are called *centers*. Our examples use the *(weighted) graph metric* in which the points are the vertices of a graph with edge lengths, and the distance between two points is the length of a shortest path between them.

## 2   Relations between proportional fairness notions

In this section, we prove the relations between proportional fairness [Chen et al., 2019], individual fairness [Jung et al., 2020], and the transferable core [Li et al., 2021]. We first define the notions.

The idea of *proportional fairness* is the following: If there is a candidate $c$ such that at least $\frac{n}{k}$ agents are closer to $c$ by a factor $\alpha$ than to their closest cluster center in the outcome $W$, then we say that the agents will *deviate* to $c$. If there is no such candidate, the outcome satisfies $\alpha$-proportional fairness.

**Definition 1.** For $\alpha \geq 1$ an outcome $W$ satisfies $\alpha$-*proportional* fairness, if there is no group $N' \subseteq N$ of agents with $|N'| \geq \frac{n}{k}$ and $c \notin W$ such that $\alpha \cdot d(i, c) < d(i, W)$ for all $i \in N'$.

While $(2 - \varepsilon)$-proportional fair outcomes need not exist (for any $\varepsilon > 0$), $(1 + \sqrt{2})$-proportional fair outcomes can be computed for any metric space [Chen et al., 2019, Micha and Shah, 2020].

To define *individual fairness*, denote by $r_{N,k}(i)$ be the radius of the smallest ball around an agent $i \in N$ that encloses at least $\frac{n}{k}$ agents, i.e., $r_{N,k}(i) = \min\{r \in \mathbb{R} \colon |B(i, r) \cap N| \geq \frac{n}{k}\}$. We drop the subscripts $N$ and $k$ if clear from context. For this definition to properly work, we additionally need the assumption that $N \subseteq C$, i.e., any agent can be chosen as center. Otherwise, a secluded group of agents without any possible cluster centers around them would never be able to get a center close to them in the outcome. Indeed, this is a plausible restriction in metric clustering, as oftentimes the centers may be picked from the (infinite) set of points in the metric space.

**Definition 2.** For an instance with $N \subseteq C$, for $\beta \geq 1$ an outcome $W$ satisfies $\beta$-*individual fairness* if $d(i, W) \leq \beta r_{N,k}(i)$ for all $i \in N$.

It is known that an outcome satisfying 2-individual fairness always exists, while there are instances with no $(2 - \varepsilon)$-individually fair outcome [Jung et al., 2020].

The *transferable core*[2] notion is based on the concept of transferable utilities from game theory. Comparing to proportional fairness, the notion considers the average utility for each group.

**Definition 3.** For $\gamma, \alpha \geq 1$, an outcome $W$ is in the $(\gamma, \alpha)$-*transferable core* if there is no group of agents $N' \subseteq N$ and candidate $c \notin W$ with $|N'| \geq \gamma \frac{n}{k}$ and $\alpha \sum_{i \in N'} d(i, c) < \sum_{i \in N'} d(i, W)$.

It is known that the for any $\gamma > 1$ there are outcomes in the $(\gamma, \max(4, \frac{3\gamma - 1}{\gamma - 1}))$-transferable core while there need not be outcomes in the $(\gamma, \min(1, \frac{1}{\gamma - 1}))$-transferable core [Li et al., 2021].

---

[2]We remark that Li et al. [2021] call this notion just "core", we rename it to avoid confusion with the core notion of Ebadian and Micha [2024].

*Example* 1. Consider the instance depicted in Figure 1 (right) with $k = 5$ and the associated graph distance metric. Assume that cluster centers can only be placed on the depicted agents. We have $\frac{n}{k} = 2$; thus any two agents are able to deviate to another center. The outcome $W = \{1, 2, 3, 6, 9\}$ satisfies 1-proportional fairness: The agents $1, \ldots, 4$ have distance 0 to a center, while every remaining agent has distance at most 1 to a center.

To see the difference between proportional fairness, individual fairness, and the transferable core, consider the same instance with $k = 4$, so $\frac{n}{k} = 2.5$. Here, the outcome $W = \{1, 2, 6, 7\}$ satisfies 1-proportional fairness, however it does not satisfy 1-individual fairness. Agent 8 could look at their 2 closest neighbors, 5 and 9, both at a distance of 1. However, the distance of 8 to the outcome is 2. Observe that $W$ also is not in the $(1, 1)$-transferable core. Here, for the group $N' = \{8, 9, 10\}$ and candidate $c = 9$, we have $\sum_{i \in N'} d(i, c) = 2 < \sum_{i \in N'} d(i, W) = 4$. ◇

## 2.1 Proportional and individual fairness

We first show that proportional and individual fairness are the same up to a factor of at most 2.

**Theorem 1.** *Let $\alpha, \beta \geq 1$. If $N \subseteq C$, then an outcome that satisfies $\alpha$-proportional fairness also satisfies $(1 + \alpha)$-individual fairness, and an outcome that satisfies $\beta$-individual fairness also satisfies $2\beta$-proportional fairness. If $N = C$, then an outcome that satisfies $\beta$-individual fairness also satisfies $(1 + \beta)$-proportional fairness.*

*Proof.* Let $W \subseteq C$ be an outcome satisfying $\alpha$-proportional fairness, $j \in N$ be any agent, and $N_j = \{i \in N : d(i, j) \leq r(j)\}$. As $N \subseteq C$, there is an $i \in N_j$ with $d(i, W) \leq \alpha d(i, j)$; otherwise the coalition $N_j$ deviates to candidate $j$. Thus, by the triangle inequality, $d(j, W) \leq d(i, j) + d(i, W) \leq (1 + \alpha)d(i, j) \leq (1 + \alpha)r(j)$, and hence $W$ satisfies $(1 + \alpha)$-individual fairness.

Now suppose the outcome $W$ satisfies $\beta$-individual fairness. Let $N' \subseteq N$ with $|N'| \geq \frac{n}{k}$ and $c \notin W$ be an unchosen candidate. Take $i^* \in N'$ to be the agent in $N'$ furthest away from $c$. If $N \subseteq C$, then the radius $r(i^*)$ containing $\lceil \frac{n}{k} \rceil$ agents is at most as large as the most distant agent in $N'$, i.e., there is an $i' \in N'$ with $r(i^*) \leq d(i^*, i') \leq d(i^*, c) + d(c, i')$. Then $d(i^*, W) \leq \beta r(i^*) \leq \beta(d(i^*, c) + d(c, i')) \leq 2\beta d(i^*, c)$. If $N = C$, then, since $|N'| \geq \frac{n}{k}$, we have $r(c) \leq d(c, i^*)$; thus $d(c, W) \leq \beta d(c, i^*)$. Therefore, $d(i^*, W) \leq d(i^*, c) + d(c, W) \leq (1 + \beta)d(i^*, c)$, and thus $W$ also satisfies $(1 + \beta)$-proportional fairness. □

Indeed, we also show that all three provided bounds are tight.

**Theorem 2.** *For every $\alpha, \beta \geq 1$ and $\varepsilon > 0$, there are instances with $N = C$ for which there exists (1) an outcome which satisfies $\alpha$-proportional fairness, but not $(1 + \alpha - \varepsilon)$-individual fairness, and (2) an outcome which satisfies $\beta$-individual fairness, but not $(1 + \beta - \varepsilon)$-proportional fairness. Moreover, there are instances with $N \subseteq C$ for which there exists (3) an outcome which satisfies $\beta$-individual fairness, but not $(2\beta - \varepsilon)$-proportional fairness.*

## 2.2 Proportional fairness and the transferable core

It is easy to see that the $(1, \alpha)$-transferable core implies $\alpha$-proportional fairness. For $\gamma > 1$ however, the $(\gamma, \alpha)$-transferable core does not imply any meaningful proportional fairness approximation (consider $\frac{n}{k}$ agents on one point and $(\gamma - 1)\frac{n}{k}$ agents "far" away). Hence, we focus on the other direction and show that a proportional fairness approximation implies one to the transferable core.

**Theorem 3.** *An outcome satisfying $\alpha$-proportional fairness is in the $\left(\gamma, \frac{\gamma(\alpha+1)}{\gamma-1}\right)$-transferable core for any $\alpha \geq 1$ and $\gamma > 1$.*

*Proof.* Let $W \subseteq C$ satisfy $\alpha$-proportional fairness. Let $N' \subseteq N$ be a group of agents of size $n' \geq \gamma \frac{n}{k}$, $c \notin W$, and shorten $\eta = \lceil \frac{n}{k} \rceil$. Further, assume the agents $N' = \{i_1, \ldots, i_{n'}\}$ are ordered by their increasing distance to $c$, i.e., $d(i_j, c) \leq d(i_{j+1}, c)$ for every $j \in [n' - 1]$. Let $J_0 = \{i_1, \ldots, i_\eta\}$ and $j_0 \in J_0$ such that $d(j_0, W) \leq \alpha d(j_0, c)$; such an agent must exist due to $\alpha$-proportional fairness. Next, for $\lambda = 1, \ldots, n' - \eta$, we inductively define $J_\lambda = \{i_1, \ldots, i_{\eta+\lambda}\} \setminus \{j_0, \ldots, j_{\lambda-1}\}$, and choose $j_\lambda \in J_\lambda$ such that $d(j_\lambda, W) \leq \alpha d(j_\lambda, c) \leq \alpha d(i_{\eta+\lambda}, c)$ (note that $|J_\lambda| = \eta$). Thus,

$$\sum_{\lambda=0}^{n'-\eta} d(j_\lambda, W) \leq \alpha \sum_{z=\eta}^{n'} d(i_z, c) \leq \alpha \sum_{i \in N'} d(i, c). \tag{1}$$

Next, for each $i \in N'' = N' \setminus \{j_0, \ldots, j_{n'-\eta}\}$, we can bound the distance to $W$ as follows:

$$d(i, W) \leq d(i, c) + d(c, j_0) + d(j_0, W) \leq d(i, c) + (1 + \alpha)d(i_\eta, c). \tag{2}$$

Note that $d(i_\eta, c) \leq \frac{1}{n' - |N''|} \sum_{z=\eta}^{n'} d(i_z, c)$ as each of the summands is at least $d(i_\eta, c)$. Thus,

$$\sum_{i \in N''} d(i, W) \leq \sum_{i \in N''} \big(d(i, c) + (1 + \alpha)d(i_\eta, c)\big) \leq (1 + \alpha)\frac{|N''|}{n' - |N''|} \sum_{z=\eta}^{n'} d(i_z, c) + \sum_{i \in N''} d(i, c). \tag{3}$$

As $n' \geq \gamma \frac{n}{k}$ and $|N''| = \eta - 1 \leq \frac{n}{k}$, we have $\frac{|N''|}{n' - |N''|} \leq \frac{1}{\gamma - 1}$. In all, $\sum_{i \in N'} d(i, W)$ is the sum of (1) and (3), which is at most

$$\left(\frac{\alpha+1}{\gamma-1} + \alpha + 1\right) \sum_{i \in N'} d(i, c) = \frac{\gamma(\alpha+1)}{\gamma-1} \sum_{i \in N'} d(i, c). \tag{4}$$

Hence, $W$ is in the $\left(\gamma, \frac{\gamma(\alpha+1)}{\gamma-1}\right)$-transferable core. $\qquad \square$

We remark that the denominator $\frac{1}{\gamma-1}$ in $\alpha$ is inevitable. This is because for $\gamma \leq 2$, the $\left(\gamma, \frac{1}{\gamma-1}\right)$-transferable core may be non-empty [Li et al., 2021, Theorem 18]. We complement the above upper bound with an asymptotically tight lower bound.

**Theorem 4.** *For any $\alpha \geq 1$, $\gamma > 1$, and $\varepsilon > 0$ there exists an instance in which an $\alpha$-proportional fair outcome is not in the $\left(\gamma, \frac{\gamma\alpha+1}{\gamma-1} - \varepsilon\right)$- transferable core.*

## 3 Fairness notions for multiwinner voting axioms

In this section we show a connection between the research on computational social choice, specifically approval-based multiwinner voting (also known as approval-based committe (ABC) voting) and the fairness notions for clustering. We will first give a primer on ABC voting and introduce our *metric JR axioms*. We then focus on two of those axioms and show that (1) they are satisfied by existing, simple algorithms, and (2) they imply the best known approximation guarantees to proportional and individual fairness, the transferable core, and the $q$-core (see Definition 5 below). For the latter two notions, we are even able to improve upon the best currently known approximation guarantees. Finally, we will focus on the case when the candidate set is infinitely large (i.e., when we are in the Euclidean space and every point is a candidate): In this setting, the above algorithms become hard to compute. We combine two approaches to maneuver around this hardness and again match upon the best known approximation guarantees for proportional fairness and the $q$-core.

### 3.1 Metric JR axioms

In ABC voting [Lackner and Skowron, 2022], we are given a set $N$ of voters (or agents), a set $C$ of candidates, and a committee size $k$. For each voter $i \in N$ we are given a subset $A_i \subseteq C$ of candidates they approve. For such preferences, we call a set $N' \subseteq N$ of voters $\ell$-*large* if $|N'| \geq \ell\frac{n}{k}$, and $\ell$-*cohesive* if $|\bigcap_{i \in N'} A_i| \geq \ell$. We say that a committee satisfies

JR if for every 1-cohesive and 1-large group $N'$ there exists an $i \in N'$ with $|A_i \cap W| \geq 1$;
PJR if for every $\ell \in [k]$ and $\ell$-cohesive and $\ell$-large group $N'$ it holds that $|\bigcup_{i \in N'} A_i \cap W| \geq \ell$,

and remark that there are many further proportionality axioms [Lackner and Skowron, 2022]. Here, JR is short for *justified representation*. To define our *metric JR axioms* for voters and candidates in a distance metric, we follow Brill and Peters [2023] (who lifted these axioms for weak ordinal preferences and called them rank-$\Pi$) and generalize their notions to look at each distance separately.

**Definition 4** (Metric JR axioms). Let $(\mathcal{X}, d)$ be a distance metric, let $N, C \subseteq \mathcal{X}$. Let $\Pi$ be a proportionality axiom. An outcome $W$ satisfies $m\Pi$ (short for metric) if for all $y \in \mathbb{R}_{\geq 0}$, for the ABC voting instance in which each $i \in N$ has the approval set $B(i, y) \cap C$, the outcome $\bar{W}$ satisfies $\Pi$.

For example, an outcome $W$ satisfies mJR if for every $y \in \mathbb{R}$ and for every group $N' \subseteq N$ of at least $\frac{n}{k}$ agents whose ball of radius $y$ all contain a common candidate ($|\bigcap_{i \in N'} B(i, y) \cap C| \geq 1$), there exists an agent $i \in N'$ whose ball of radius $y$ contains a center $c \in W$.

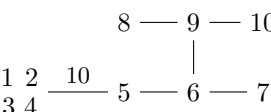

Figure 2: Metric space for some of the examples. Edges without labels have length 1.

We want to point out that mJR is *significantly* weaker than mPJR. Indeed, to satisfy mPJR, an outcome $W$ may need to contain several candidates $c$ such that $d(i, c) > d(i, W)$ for all agents $i \in N$, i.e., $c$ is no-ones "first choice" among $W$; mJR does not have this property. This makes mJR the more sensible of the two axioms for "vanilla" clustering, in which one only cares about the closest center to each agent. mPJR however, is a natural axiomatic choice for settings such as sortition or even social choice in general: Here, agents may benefit from having more than a single representative. We provide some intuition for mJR and mPJR and this property in the example below.

*Example* 2. To see the differences between the proportionality axioms, consider the instance depicted in Figure 1 (right). First consider instance (a) on the left with $N = C = \{1, \ldots, 10\}$, $k = 4$, and the outcome $W = \{1, 2, 3, 6\}$. Here, $\frac{n}{k} = 2.5$. First, we note that this outcome does not satisfy 1-proportional fairness: The agents $8, 9, 10$ are closer to 9 than they are to the closest winner in $W$. It does however satisfy mJR: Among every group of at least three agents that have a common candidate within distance $y$, there is one agent that has a cluster center $w \in W$ within distance $y$. For example, $8, 9, 10$ have candidate 9 at distance 1, and the distance of 9 to the closest center is also 1. This outcome does not satisfy mPJR though, since the group $5, \ldots, 10$ would deserve at least two candidates within distance 1 in $W$. An outcome satisfying mPJR is $W = \{1, 2, 3, 9\}$. For $y = 0$, only the group $\{1, \ldots, 4\}$ shares a candidate, but also have a center at distance 0.

If $k = 5$, then, to satisfy mPJR, an outcome must contain at least two of the four candidates. But there are outcomes satisfying mJR that contain only one of $1, \ldots, 4$. This property of mPJR makes it suited for settings in which agents may want to be represented by multiple candidates, e.g., in political settings, in which the candidates end up possessing voting power to represent the agents. ◇

Independently of Brill and Peters [2023], Aziz et al. [2024] introduced two notions they call *Proportionally Representative Fairness*. The first notion is called "discrete" (DPRF), and the second is called "unconstrained" (UPRF). Indeed, DPRF is equivalent to mPJR. UPRF was introduced to tackle the case when the candidate space is unbounded. We discuss how it relates to mPJR and the other fairness notions in the full version [Kellerhals and Peters, 2023].

Aziz et al. [2024] show that an outcome satisfying DPRF (mPJR) also fulfills $\left(1 + \sqrt{2}\right)$-proportional fairness. We show hereafter that this already holds for the (much weaker) mJR axiom.

## 3.2 Fairness bounds for mJR outcomes

We now prove the approximation guarantees implied by mJR. We remark that the bound for the transferable core below improves upon the analysis of Li et al. [2021]. The proof is deferred to the full version [Kellerhals and Peters, 2023].

**Theorem 5.** *Let $W$ be an outcome satisfying mJR. Then it also satisfies $\left(1 + \sqrt{2}\right)$-proportional fairness, 2-individual fairness, and is in the $\left(\gamma, \frac{2\gamma}{\gamma-1}\right)$-transferable core for any $\gamma > 1$.*

If the candidate space is finite, then an outcome satisfying mJR can be computed in polynomial time by the GREEDY CAPTURE algorithm [Chen et al., 2019, Micha and Shah, 2020, Li et al., 2021]. We briefly recall its procedure:

GREEDY CAPTURE starts off with an empty clustering $W$. It maintains a radius $\delta$ (initially $\delta = 0$) and smoothly increases $\delta$. If there is a candidate $c$ such that at least $\frac{n}{k}$ agents have distance at most $\delta$ to $c$, it adds $c$ to $W$ and deletes the $\frac{n}{k}$ agents. If an agent has distance at most $\delta$ to a candidate in $W$, then it is deleted as well. This is continued until all agents are deleted.

*Example* 3. Consider the instance in Figure 2. Here, with $k = 4$, GREEDY CAPTURE, would first open one of $1, \ldots, 4$ with $\delta = 0$ and remove all agents from $1, \ldots, 4$. Then for $\delta = 1$ it could either open 6 or 9, removing all adjacent agents to it. Then there are two agents remaining, which would

be assigned to either 6 or 9 for $\delta = 2$. Thus, in this instance, GREEDY CAPTURE only opens two clusters. $\diamond$

**Proposition 6.** *Any outcome returned by* GREEDY CAPTURE *satisfies mJR.*

### 3.3 Fairness bounds for mPJR outcomes

Recall that mPJR is equivalent to the DPRF notion by Aziz et al. [2024]. To satisfy their notion, they designed a generalization of the expanding approvals rule from multiwinner voting [Aziz and Lee, 2020, 2021] (in which the agents' preferences over the candidates are ordinal) to the setting of proportional clustering. They refer to this generalization as SPATIAL EXPANDING APPROVALS. As Aziz et al. show, SPATIAL EXPANDING APPROVALS can be computed in polynomial time for finite candidate spaces.

In general, SPATIAL EXPANDING APPROVALS behaves similarly to GREEDY CAPTURE. It also starts off with an empty clustering $W$ and a radius $\delta = 0$ as well and additionally gives each agent a budget $b_i = \frac{k}{n}$. It then smoothly increases the radius $\delta$. When there is a candidate $c \notin W$ for which the agents at a distance of at most $\delta$ have a budget of at least 1, it decreases the budget of these agents collectively by exactly 1 and adds $c$ to $W$.

*Example* 4. Consider the instance in Figure 2. Here, with $k = 4$, SPATIAL EXPANDING APPROVALS would give each agent a budget of $\frac{4}{10} = \frac{2}{5}$. For $\delta = 0$, it will open a cluster from $1, \ldots, 4$ and decrease their budgets by exactly 1, for instance it could set the budget of 1 and 2 to 0 and of 3 to $\frac{1}{5}$. Then for $\delta = 1$ it could again open 6 and 9, for instance by removing the budget of 6 and 9 to $\frac{1}{5}$ and of $5, 7, 8, 10$ to zero. The remaining budget is exactly 1, which would be spent for $\delta = 10$ on 5. Thus, one possible final outcome is $\{1, 5, 6, 9\}$. $\diamond$

Remarkably, as shown by Aziz et al. [2024], it does not matter in which way the budget is subtracted and which candidate meeting the budget is selected; the outcomes of the algorithm will satisfy mPJR in any case.

**Proposition 7.** *Any outcome returned by* SPATIAL EXPANDING APPROVALS *satisfies mPJR.*

We now turn to fairness measures implied by mPJR. As any outcome satisfying mPJR also fulfills mJR, the results in Theorem 5 also hold for mPJR. Indeed, mPJR is stricter in the sense that larger groups must also be represented justly by a proportional number of candidates: an $\alpha$ percentage of the population should roughly be close to an $\alpha$ percentage of the centers.

This property makes mPJR fit well into metric social choice settings such as sortition. For this, Ebadian and Micha [2024] introduced a fairness notion that measures proportionality in this setting by considering for each agent not only the closest center, but the first $q$ closest centers. In that, their notion called $\alpha$-$q$-core naturally generalizes $\alpha$-proportional fairness; the two are equal when $q = 1$.

**Definition 5.** For $\alpha \geq 1$ an outcome $W$ is in the $\alpha$-$q$-core, if there is no $\ell \in \mathbb{N}$ and no $N' \subseteq N$ with $|N'| \geq \ell \frac{n}{k}$ and set $C' \subseteq C$ with $q \leq |C'| \leq \ell$ such that $\alpha \cdot d^q(i, C') < d^q(i, W)$ for all $i \in N'$.

*Example* 5. Consider the instance in Figure 2 with $k = 5$ and the outcome $W = \{1, 2, 3, 6, 9\}$. As mentioned above, $W$ satisfies 1-proportional fairness. For the 3-core however, consider the set $N' = \{5, \ldots, 10\}$ deviating to $C' = \{6, 9, 10\}$. The distance of any member of $N'$ to 6, 9, or 10 is at most 3, while the distance to the third most distant center in the outcome is at least 10. Thus, when considering the distances to the third most distant candidate in $C'$, every agent in $N'$ would improve by at least a factor of $\frac{10}{3}$. Thus, $W$ is only in the $\frac{10}{3}$-3-core. $\diamond$

We mention in passing that we introduce similar gerneralizations for individual fairness and the transferable core, in which each agent is represented by $q$ candidates instead of one. The definitions and obtained results can be found in the full version of this paper [Kellerhals and Peters, 2023].

Ebadian and Micha [2024] show that, if $N = C$ (every agent is a candidate and vice versa), for a given $q$, one can compute a $\frac{5+\sqrt{41}}{2}$-$q$-core outcome.[3] We show that mPJR (or DPRF) provides a better guarantee for the $q$-core, for all values of $q$ simultaneously.

**Theorem 8.** *If an outcome satisfies mPJR, then, for every $q \leq k$, it is in the 5-$q$-core.*

---

[3]In addition, their randomized algorithm selects each agent with the same probability, a desirable property in the context of sortition, i.e., the randomized selection of citizen assemblies [Flanigan et al., 2021].

To prove the theorem, we use two lemmas. The first was first observed by Ebadian and Micha [2024, Lemma 1] and is proven here in a shorter fashion.

**Lemma 9.** *Let $\ell \geq q \geq 0$, let $N' \subseteq N$ be a set of agents with $N' \geq \ell \frac{n}{k}$, and let $C' \subseteq C$ be a set of $q \leq |C'| \leq \ell$ candidates such that $d^q(i, C') \leq d^q(i, W)$ for any $i \in N'$. Then there is a set $N'' \subseteq N$ of at least $q\frac{n}{k}$ agents and a candidate $c \in C'$ such that $d(i, c) \leq d^q(i, C')$ for all $i \in N''$.*

*Proof.* Assume that each agent marks each of their top $q$ choices among $C'$. Then there are at least $q|C'|\frac{n}{k}$ marks on the candidates. Thus, there is one $c \in C'$ with at least $q\frac{n}{k}$ marks. □

The next lemma bounds the $\alpha$-$q$-core once we find two agents with specific bounds on their distances.

**Lemma 10.** *Let $\rho_1, \rho_2, \sigma_1, \sigma_2 \geq 0$ and let $W \subseteq C$ be an outcome. If for any set $N' \subseteq N$ of at least $\ell \frac{n}{k}$ agents and any candidate set $C' \subseteq C$ with $q \leq |C'| \leq \ell$ there are $i_1, i_2 \in N'$ such that $d^q(i_1, W) \leq \rho_1 d^q(i_1, C') + \rho_2 d^q(i_2, C')$ and $d^q(i_2, W) \leq \sigma_1 d^q(i_1, C') + \sigma_2 d^q(i_2, C')$, then $W$ is in the $\alpha$-$q$-core, where $\alpha \leq \rho_1 + \frac{1}{2}\left(\sigma_2 - \rho_1 + \sqrt{(\rho_1 - \sigma_2)^2 + 4\sigma_1\rho_2}\right)$.*

*Proof of Theorem 8.* Let $N' \subseteq N$ be a group of agents with $|N'| \geq \ell \frac{n}{k}$ and let $C' \subseteq C$ with $q \leq |C'| \leq \ell$ such that $d^q(i, C') \leq d^q(i, W)$ for any $i \in N'$. By Lemma 9 there is a candidate $c$ being ranked in their top $q$ among $C'$ by $q\frac{n}{k}$ many agents $N'' \subseteq N'$. Out of $N''$, let $i_1$ be the agent maximizing $d(i_1, c)$ and $i_2$ be any other agent in $N''$. Also, let $C'' \subseteq C'$ be the set of the $q$ candidates closest to $i_2$. Then, for every $c' \in C''$ and every $i \in N''$, we have

$$d(i, c') \leq d(i, c) + d(c, i_2) + d(i_2, c') \leq d(i_1, c) + 2d^q(i_2, C') =: y.$$

In other words, $|\bigcap_{i \in N''} B(i, y) \cap C| \geq q$. Now mPJR implies that $|\bigcup_{i \in N''} B(i, y) \cap W| \geq q$. Thus, for every $i \in N''$ there is an agent $i' \in N''$ such that $d^q(i, W) \leq d(i, i') + y$. Since the distance of $i_1$ to any other agent $i' \in N''$ is $d(i_1, i') \leq d(i_1, c) + d(c, i_1)$, we have

$$d^q(i_1, W) \leq 2d(i_1, c) + y \leq 3d^q(i_1, C') + 2d^q(i_2, C').$$

As the distance of $i_2$ to any other agent $i' \in N''$ is at most $d(i', c) + d(c, i_2) \leq d(i_1, c) + d^q(i_2, C')$,

$$d^q(i_2, W) \leq d(i_1, c) + d^q(i_2, C') + y \leq 2d(i_1, c) + 3d^q(i_2, C') \leq 2d^q(i_1, C') + 3d^q(i_2, C').$$

Applying Lemma 10 with $\rho_1 = \sigma_2 = 3$ and $\rho_2 = \sigma_1 = 2$ yields the stated 5-$q$-core. □

### 3.4 Dealing with unbounded candidate sets

Whenever the candidate space $C$ is finite, it is straightforward to implement GREEDY CAPTURE and SPATIAL EXPANDING APPROVALS in polynomial time. However, as shown by Micha and Shah [2020], once $C$ is unbounded and the metric space is only implicitly given (e.g., some distance norm over $C = \mathbb{R}^t$), computing GREEDY CAPTURE can become NP-hard. For Euclidean distances over $C = \mathbb{R}^t$, Micha and Shah [2020, Theorem 12] were nevertheless able to give an approximate version of GREEDY CAPTURE, which approximates proportional fairness up to a factor of $2 + \varepsilon$ for any $\varepsilon > 0$ in this special metric space. For general metric spaces, Micha and Shah [2020, Theorem 11] show that in an instance with $N \subseteq C$, any outcome which is $\alpha$-proportionally fair when restricted to the instance with candidate set $N$ is $2\alpha$-proportionally fair in the whole instance. Aziz et al. [2024] used a very similar approach to this and showed that running SPATIAL EXPANDING APPROVALS on the agents results in a 3-proportionally fair outcome. Combining both approaches, we show that any outcome $W$ satisfying mJR when restricted to the instance with candidate set $N \cup W$ satisfies 3-proportional fairness in the entire instance. Thus, GREEDY CAPTURE restricted to the agents yields a 3-proportionally fair outcome. The same also applies to mPJR and the $q$-core.

**Theorem 11.** *Consider an instance $I$ with $N \subseteq C$ and an outcome $W$ and let $I'$ be the instance with agent set $N$ and candidate set $N \cup W$. If $W$ satisfies mJR in $I'$, then $W$ satisfies 3-proportional fairness. If $W$ satisfies mPJR in $I'$, then $W$ is in the 4-$q$-core for all $q \leq k$.*

# 4 Stronger fairness bounds for sortition

Ebadian and Micha [2024] introduced FAIR GREEDY CAPTURE, a randomized generalization of GREEDY CAPTURE for the setting of sortition. It works in the setting in which $N = C$ and is parameterized by some parameter $q \leq k$. Like GREEDY CAPTURE it smoothly increases a radius around each agent/candidate. Once this radius contains at least $q\frac{n}{k}$ agents, it selects $q$ of them uniformly at random and deletes in total $\lceil q\frac{n}{k} \rceil$ of these agents. Together with an adequate final sampling step, one can show that this selects each agent with a probability of exactly $\frac{k}{n}$.

Ebadian and Micha show that any clustering returned by the algorithm is in the $\frac{3+\sqrt{17}}{2}$-1-core when parameterized by $q = 1$ and in the $\frac{5+\sqrt{41}}{2} \approx 5.7$-$q$-core when parameterized by $q > 1$.[4] We improve upon their analysis (with a simpler proof) and show that FAIR GREEDY CAPTURE satisfies a better bound for every parameter $q \leq k$.

**Theorem 12.** *Let $N = C$ and $q \leq k$. Then any outcome $W$ returned by* FAIR GREEDY CAPTURE *parameterized by $q$ is in the $\frac{3+\sqrt{17}}{2}$-$q$-core.*

# 5 Conclusion and future work

In this paper, we studied proportional clustering from a social choice perspective and showed that our new *metric JR axioms* enable near-optimal approximations of fairness notions for clustering. An interesting open question, both relevant to social choice and clustering is related to a different relaxation of proportional fairness (or core fairness) introduced by Jiang et al. [2020]. Instead of bounding the factor by which the agents can improve, they bound the size of the deviating coalition (similar to the transferable core). In that sense, no group of size $\gamma\frac{n}{k}$ should exist, who could all deviate to a candidate they like more. In their work, they show that there are instances for which no solution with $\gamma < 2$ can exist while for any $\varepsilon > 0$ a solution with $\gamma = 16 + \varepsilon$ exists. Since these results only care about the relative ordering of the candidates, they also translate to clustering. Closing this bound, or improving it for certain metric spaces, seems like an interesting problem. It would be also intriguing to study the probabilistic analog of the core [Cheng et al., 2020, Jiang et al., 2020], especially if the results generalize to the $q$-core and if certain metric spaces admit simple algorithms to compute it.

Further, SPATIAL EXPANDING APPROVALS (Section 3.3) is more of a family of algorithms, parameterized by how candidates are selected and how budgets are deducted. Is there any way to axiomatically (or quantitatively) distinguish its different parameterizations? In the context of approval-based multi-winner voting, the *Method of Equal Shares* [Peters and Skowron, 2020] can be seen as an instantiation of SPATIAL EXPANDING APPROVALS which provides stronger proportionality guarantees than other algorithms in the family. Is something similar possible for our setting, e.g., can one go from proportionality axioms inspired by PJR to axioms inspired by the stronger EJR axiom [Aziz et al., 2017]? As shown by Brill and Peters [2023, Example 7] the straightforward extension of studying mEJR (or rank-EJR in their notation) is not possible, as outcomes satisfying mEJR may easily fail to exist. However, the metric variant of the PJR+ axiom of Brill and Peters [2023] may be of greater interest in the clustering setting. It is easy to see that SPATIAL EXPANDING APPROVALS satisfies mPJR+. Is it also possible to derive better proportionality or core approximations from mPJR+?

Naturally, our work still leaves several open questions when it comes to the approximation factors of our notions. What are the best attainable factors for proportional fairness and the $q$-core? Further, the questions of Jung et al. [2020] whether the bound of 2 on individual fairness can be improved for Euclidean spaces and of Micha and Shah [2020] whether for (unweighted) graph metrics with $N = C$ a 1-proportional fair clustering always exist, are still open.

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
