# OpenReview forum: "Proportional Fairness in Clustering: A Social Choice Perspective"
_NeurIPS.cc/2024/Conference — NeurIPS 2024 poster_

### Official Review · Reviewer_qn3s · 2024-07-11

**Soundness:** 4
**Presentation:** 3
**Contribution:** 2
**Rating:** 6
**Confidence:** 3

**Summary:**

This paper studies three different related notions of fair clustering that have been proposed in the literature for metric centroid clustering: (1) proportional fairness, (2) individual fairness, and (3) transferable core. Prior work has separately defined, motivated, and studied the existence and computation of clusterings optimizing all of these notions of fairness.

The present work attempts to synthesize these different notions in two senses. First, the paper shows that up to [small] constant factors, approximate proportional fairness implies approximate individual fairness and vice versa, and approximate proportional fairness implies approximate transferable core.

Second, the paper considers metric adaptations of the concepts of justified representation (mJR) and proportional justified representation (mPJR) from approval-based committee selection, an important fair representation problem in computational social choice, and show that these imply approximations of the aforementioned fair clustering concepts. Specifically, any solution satisfying mJR simultaneously satisfies the best known approximations to individual and proportional fairness, and to the transferable core. The paper shows that the Greedy Capture algorithm from prior work on proportional fairness satisfies this mJR property and thus is simultaneously approximately fair in all three senses. Furthermore, the Expanding Approvals algorithm from prior work satisfies the stronger mPJR property. The paper turns to the implications of mPJR for another generalized notion of proportional fairness (the q-core) in contexts where agents measure cost by distance to the q’th most distant center (q=1 being the typical centroid clustering setting) motivated by prior work in sortition (selecting a fair and representative sample of a population), showing that mPJR implies a 5-approximation for all q.

Finally, the paper shows two additional results extending prior work. First, that the known efficient algorithms (e.g., Greedy Capture and Expanding Approvals) can be extended to handle the case of an infinite number of center candidates (e.g., can select any points in the real plane) by simply considering the set of agents as the candidates, and that this restriction preserves small constant approximations of fairness. Second, that the analysis of the Fair Greedy Capture algorithm from recent work on sortition can be tightened to yield a better constant bound on the approximation of the q-core.

**Strengths:**

The paper does a very good job of providing a comprehensive survey connecting different fair clustering desiderata that have been considered separately in the literature. The result is a convincing story that there is a more “general” phenomenon of proportionality at work in all these group-agnostic structural notions of fairness in clustering. The greatest contribution may therefore be in clarifying the state of the field for future work seeking to extend or build on these notions. It is also nice to see a formal connection between the last several years of work on fair approval-based committee selection in computational social choice and the work in the ML clustering space.

Generally the writing and discussion is clear. Effort has clearly been put into communicating ideas effectively in the forms of the results diagram in figure 1, illustrative examples, and extensive consideration of related work and the connections between other work and the current.

Results are well substantiated by the formal arguments, and claims are all measured and accurate to what is shown.

**Weaknesses:**

The work is largely incremental within the fair clustering space, focusing on the relationship between prior notions of fairness, the existence and computation of which have already been studied. The algorithmic contribution is limited in this sense as well, focusing mostly on tightening or demonstrating relationships between guarantees and analysis. In these senses, the impact of the paper may be limited.

Post-rebuttal clarification to the above: The weakness discussed is really with respect to algorithmic contributions, and I agree that connecting different work in the field is a valuable contribution.

A minor comment: In Lemma 9, lines 315-316, I think it should be “…let C’ be a set of…candidates” and “…there is a set $N^{''} \subseteq N^{'}$...”

**Questions:**

No pressing questions, though if the authors feel that I have misunderstood the potential of the work for improving clustering algorithms in my weaknesses, I would be happy to hear different perspectives.

**Limitations:**

No concerns.

---

> ### Author Rebuttal · Authors · 2024-08-07
>
> Thank you kindly for the minor comment, you are absolutely correct.
> We agree that our work has a clear focus on creating relations between the fairness measures.
> However, we believe that providing connections and (in-)compatibility between fairness notions is of great importance, and something that has not been considered enough.
> As we mention in the introduction, this belief is also shared by Dickerson et al. [2023b]; (their manuscript since has been published on arXiv: https://arxiv.org/abs/2406.15960).
> Also, with the connection to multiwinner voting, we create a new pathway for future work on clustering with fairness measures a la individual or proportional fairness.

---

> > ### Comment · Reviewer_qn3s · 2024-08-10
> >
> > I acknowledge that I have read the author rebuttal. I appreciate the perspective and agree that the work contributes a helpful connection between different considerations of fairness notions.

---

### Official Review · Reviewer_g3Lo · 2024-07-12

**Soundness:** 4
**Presentation:** 4
**Contribution:** 3
**Rating:** 8
**Confidence:** 3

**Summary:**

The paper unifies several definitions of fairness in clustering and relates them through approximation ratios. The paper provides theoretical results that 'translate' a definition into another, providing tight bounds for proportional and individual fairness metrics. The paper concludes with a series of results derived from multiwinner voting theory that relate fairness to justified representation.

**Strengths:**

The paper is very-well written, with substantial theoretical contributions. The relationship with prior work is comprehensive and clear, and it does a great job of extracting methods in order to relate and unify different definitions of fairness. The problem posed is important, as fairness in clustering may require different definitions, and as such, knowing how they relate to each other provides insight into the potential trade-offs. I appreciate the examples that show tight bounds for individual and proportional fairness.

**Weaknesses:**

It would be worth discussing the asymmetry in some of the theoretical results (e.g. individual and proportional fairness relate asymmetrically). The alpha-PF -> (gamma, alpha)-TC edge visualization in Fig 1 is slightly confusing: would be good to mention that the gamma(alpha + 1)/(gamma-1) is about relating the alphas.

**Questions:**

1. It would be good to discuss the problem of clustering when the cluster centers are not part of the data but are a la k-means algorithms, found from some continuous space: are any of the results or methods transferable?

2. It would be great to discuss applications of the work in the context of fair clustering, especially since multi-winner voting is mentioned: what decision-making processes would entail using one vs. more fairness definitions, and how would potential trade-offs derived from the approximation bounds affect the outcome on the population?

**Limitations:**

It would be great to include a more comprehensive discussion on limitations.

---

> ### Author Rebuttal · Authors · 2024-08-07
>
> 1. Yes, we discuss this in section 3.4. Recall that the agent set N corresponds to the point set in k-means, and the centers are chosen from the candidate set C.
> In section 3.4, we discuss results for the case that the candidate set is infinite. This also covers e.g. the case when the centers may be chosen from all of the Euclidean space.
>
> 2. Generally speaking, fairness notions such as individuality or proportionality are efforts towards modeling what is considered fair. Our work shows that these notions are highly related to another. That is, if my goal is to have a proportionally fair solution, then I implicitly also obtain a solution that is (approximately) individually fair.
> Considering for example any solution that satisfies mJR, it fulfills the best known approximations to individual and proportional fairness at the same time, so there is not really any trade-off that we are aware of.
> We hope that this roughly answers your question. Of course, we are happy to further discuss this in the discussion period.
>
> Also, thank you for the weaknesses you pointed out. We will make sure to improve on these two points in the next version.

---

> > ### Comment · Reviewer_g3Lo · 2024-08-13
> >
> > Thanks for the answers!

---

### Official Review · Reviewer_M2yM · 2024-07-14

**Soundness:** 4
**Presentation:** 3
**Contribution:** 4
**Rating:** 7
**Confidence:** 4

**Summary:**

The authors study the setting of clustering problem where voters and candidates lie in the metric space and the goal is to elect k candidates representing groups of voters. The authors show interesting connections between this setting and the setting of proportional approval-based committee elections, i.e., they show that the known proportionality axioms can be easily adapted to this setting, yielding the best possible approximations of the notions of proportional fairness and individual fairness defined for the clustering model. In their work, the authors also study the connections between proportional fairness and individual fairness, and analyze the sortition problem.

**Strengths:**

This is a very good paper. The idea of metric variants of justified representation axioms is interesting and novel - I believe it has a potential to be further studied in follow-up works. The paper is clearly written and the results are technically sound.

**Weaknesses:**

I was a bit surprised that the authors chose to extend the axioms of JR and PJR to their setting, skipping the stronger axioms of EJR, EJR+ and PJR+ mentioned in the same paper of Brill and Peters 2023 that they cite. I think this decision should be explained in the text. But I do not see this as a major weakness of the paper.

**Questions:**

1. Do you think that the axioms of EJR, EJR+ and PJR+ can be adapted to your setting as well?

**Limitations:**

The authors adequately addressed the limitations and there are no potential negative societal impact of their work.

---

> ### Author Rebuttal · Authors · 2024-08-07
>
> Thank you for pointing this out. Of course, this deserves a discussion in the conclusion, and we want to add this in the next version.
> Indeed, the Expanding Approvals algorithm satisfies the stronger mPJR+.
> However, we were not able to show mPJR+ implies stronger bounds than mPJR.
> As for EJR and EJR+, it follows from Brill and Peters [2023] that these do not always exist.
> Hence, we chose to focus on mJR and mPJR. We also believe that it is interesting that already these weak axioms imply the best approximations to individual and proportional fairness, and to the transferable core.

---

> > ### Comment · Reviewer_M2yM · 2024-08-13
> >
> > Thank you for the response. Indeed it would be good to discuss this in conclusion.

---

### Official Review · Reviewer_e2hb · 2024-07-16

**Soundness:** 3
**Presentation:** 3
**Contribution:** 3
**Rating:** 6
**Confidence:** 3

**Summary:**

This paper provides a methodological contribution by bridging three different notions of fairness in clustering: Individual (where every agent is assigned a cluster center no farther from the $\frac{n}{k}$ neighbor), proportional fairness (no group of size $\geq$ $\frac{n}{k}$ should be able to propose a center that would improve their situation collectively),  and core-fairness (k-clustering is in the core if no group containing $\geq \frac{n}{k}$ agents can strictly decrease their total distance by deviating to a new center). The paper shows that any approximation to proportional fairness is also an approximation to individual fairness and core-fairness and vice-versa. The paper also draws connections to multi-winner voting from computational social science and reinterprets the proportionality fairness notions, leading to efficiently computable metrics (distance-based) under some constraints.

**Strengths:**

+ The paper tackles an interesting topic

+ The paper is well-written, and it makes a significant methodological contribution

+ The methodology is adequately sound and well-explained. Particularly, the connections to multi-winner settings are novel.

**Weaknesses:**

I am mostly satisfied with the paper, and I don't see any major weaknesses. It's a decent methodological contribution.

**Questions:**

I don't have any specific questions for the authors.

**Limitations:**

yes

---

### Author Rebuttal · Authors · 2024-08-07

We would like to express our gratitude to the reviewers for taking the time to carefully review our submission.
We address each of the reviewer's comments and questions below and look forward to further discussions in the coming days.

---

### Decision · Program_Chairs · 2024-09-25

**Decision:**

Accept (poster)

**Comment:**

The authors establish interesting connections between different notions of fairness in clustering and also relate this notion to the area of multi-winner voting in computational social choice. All the reviewers agreed that the technical contributions are novel, and some results e.g. the metric variants of axioms have potential for follow-up works. Although the machine learning community mainly studies statistical and algorithmic aspects of fairness, I believe a paper on the axiomatic framework and connections can be useful to the community. Therefore, in addition to the changes suggested by the reviewers, I encourage the authors to discuss/motivate further the implications of various axioms (e.g. mJR or mPJR) on the theory and practice of machine learning fairness.